# Feasibility of Reduced Ingot Hot-Top Height for the Cost-Effective Forging of Heavy Steel Ingots

**DOI:** 10.3390/ma13132916

**Published:** 2020-06-29

**Authors:** Nam Yong Kim, Dae-Cheol Ko, Yangjin Kim, Sang Wook Han, Il Yeong Oh, Young Hoon Moon

**Affiliations:** 1Research and Development Center, Taewoong Co., Ltd., Noksansandan 27-ro, Gangseo-gu, Busan 46751, Korea; namyong.kim@taewoong.com; 2Graduate School of Convergence Science, Pusan National University, Busandaehak-ro 63, Geumjeong-gu, Busan 46241, Korea; dcko@pusan.ac.kr; 3School of Mechanical Engineering, Pusan National University, Busandaehak-ro 63, Geumjeong-gu, Busan 46241, Korea; yangjin@pusan.ac.kr (Y.K.); swhan@pusan.ac.kr (S.W.H.); iyoh@pusan.ac.kr (I.Y.O.)

**Keywords:** forging, heavy ingot, hot top, shrinkage porosity, main shaft, bar

## Abstract

Feasibility studies have been performed on ingots with reduced hot-top heights for the cost-effective hot forging of heavy ingots. The quality of the heavy ingots is generally affected by internal voids, which have been known to be accompanied by inclusions and segregation. To guarantee the expected mechanical performance of the forged products, these voids should be closed and eliminated during the hot open die forging process. Hence, to effectively control the internal voids, the optimum hot-top height and forging schedules need to be determined. In order to improve the utilization ratio of ingots, the ingot hot-top height needs to be minimized. To investigate the effect of the reduced hot-top height on the forged products, shaft and bar products have been manufactured via hot forging of ingots having various hot-top heights. From the operational results, the present work suggests effective forging processes to produce acceptable shaft and bar products using ingots having reduced hot tops. The mechanical properties of shop-floor products manufactured from ingots with reduced hot tops have also been measured and compared with those of conventional ingot products.

## 1. Introduction

Following the rising demands of production efficiency and security, an increasing number of large forgings are applied in various fields including heavy industrial machinery. However, large forgings are often afflicted by process-induced internal void defects due to the inevitable non-uniform solidification of large ingots during casting. Internal voids in steel ingots significantly deteriorate the mechanical properties of the forged products and may nucleate a crack or become a source for defects during the subsequent forging steps [1,2,3,4,5]. During the forging operation, the voids present in cast ingots should be eliminated to guarantee the mechanical performance of the forged products. Numerous previous studies report on a multitude of process design approaches and on the modeling and numerical analysis of the effects of design parameters on the closure of internal voids and metal flow in the bulk forming processes [6,7,8,9,10,11,12]. Kim et al. [6] developed a forging pass schedule algorithms for billet to square bar using a neural network. The algorithms based on the void closure in forged products provide reasonable reliability when applied in industrial environments. Lee et al. [7] investigated internal void closure during the forging of large cast ingots using a simulation approach. Through the comparison of experimental results and numerical simulation, a criterion for void closure based on the local effective strain was proposed. Internal voids and non-metallic inclusions are detrimental in the performance of steel in structural and mechanical applications [8]. To reduce the danger of internal defects, the control of metal flow during deformation would be very helpful. The metal flow during bulk forming is different from that of sheet metal forming [9,10], and an analytical model that can provide a reliable guide for the metal flow would be very helpful. In order to deliver safe and sound products without internal defects, the metal flow during hot deformation for a given die geometry is very important. To minimize the internal defects during forming, Gordon et al. [11,12] proposed a reasonable solution that can be used to determine streamlined die shapes for sound products.

To control the internal voids, both the ingot hot-top height and forging schedules need to be optimized. Nevertheless, previous works dealing with ingot hot-tops have been mainly focused on the characterization and determination of the optimum ingot hot-top height. Flemings [13] has conducted studies on hot-top design based on the heat and mass flow theory and proposed guidelines for ideal hot-top design. Subsequently, Tashiro et al. [14] experimentally investigated the design conditions of the hot tops. They showed that the thermal conditions of the hot tops have little impact on the solidification behaviors of the ingot body compared to the ingot’s geometry. Scepi et al. [15] reported that the weight percent of hot tops should be higher than 23% to improve the quality of heavy ingots. However, these suggestions conflict with the ideal hot top design guidelines suggested by Flemings. Kermanpur et al. [16] investigated the proper dimension of hot tops via numerical simulations. Dang et al. [17] numerically analyzed the influence of the hot-top height on the inner quality of large forging ingots. Ma et al. [18] analyzed the influence of the hot-top heat preservation characteristics on the solidification behavior of steel ingots. 

Despite the considerable amount of work reported on the characterization and determination of optimum ingot hot-top height, no attempts have been made yet to apply ingot hot-top height reduction to improve the utilization ratio of ingots for more cost-effective hot forging.

Conventionally, hot tops are cut off to ensure the inner quality of the forged products. However, in terms of improving material yield ratio and energy efficiency, reduced hot tops are expected to gain popularity among most companies [19,20,21,22]. Thus, the rising possibility of internal defects generated in the ingots due to the reduced hot tops should receive extensive considerations. 

In order to improve the utilization ratio of ingots, the ingot hot-top height needs to be reduced. It is worth noting that the proper combination of forging schedules with reduced hot tops can overcome the creation of void defects. To develop tailored forging schedules for reduced ingot hot-top heights, various hot forging schedules have been implemented and compared for ingots of different hot-top heights. The present work suggests effective forging processes that produce acceptable shaft and bar products using ingots of reduced hot tops. The mechanical properties of shop floor product using ingots of reduced hot tops are also measured and compared with those produced using conventional ingots. 

To investigate the effect of the reduced hot-top height on the forged products, shaft and bar products have been manufactured via hot forging of ingots having various hot-top heights. The experimental study of ingot casting is not always possible and appropriate due to the high variability of the experimental parameters and environmental factors. Thus, computational simulation is a feasible alternative owing to the reduced cost and shorter analysis time. [23,24,25,26,27,28,29].

Several attempts have already been made to understand the solidification characteristics of heavy ingots through numerical simulations of ingot casting processes using the finite element method (FEM) [16,30,31,32,33,34]. Hence, to investigate the effects of the reduced hot-top height on the casting qualities of ingots, the ingot casting operations were simulated by the THERCAST^®^ 3D finite-element code. These simulations can elucidate the ingot solidification conditions in order to determine the quality of the cast ingots. 

## 2. Research Methodology

### 2.1. Finite Element Analysis of Ingot Casting

To investigate the effects of the reduced hot-top height, the casting of ingots with various hot-top heights has been compared. The diameter of the compared hot tops was 1151 mm, and the diameters of their upper and lower bodies were 1251 and 1111 mm, respectively. A schematic drawing of the Cr-Mo ingots used to manufacture the main shafts is shown in Figure 1. The heights of the three investigated hot tops were 320, 144, and 0 mm (free hot top), respectively. The heights of the hot tops were chosen so that their weight corresponds to 16% (conventional hot top), 8%, and 0% (free hot top) of the total ingot weight (Table 1).

The steel melt was bottom poured at 1531 °C (that is 30 °C higher than the liquidus temperature of Cr-Mo steel) at a flow rate of 160 mm/min through a gate with a 70 mm diameter. The ingot was surrounded by the cast steel mold with a chemical composition of 3.8C-1.0Si-0.4Mn and insulation materials. The whole mold cooled by natural air convection. 

For the model formulation in THERCAST^®^ 3D (Transvalor, Biot, France), the process was modelled as a bottom-pouring uphill casting. For the casting of steel ingots, the most important technological boundary conditions include the casting temperature and the heat equation for each of the subdomains, such as convection, imposed temperature, interface heat exchanges, etc. The heat boundary conditions allow defining of the heat transfer between the outside faces of a given domain and the air. In this study, the heat losses were defined using the emissivity, external temperature, and a heat transfer coefficient.

The thermal equation is based on the resolution of the heat transfer equation, which is a general energy conservation Equation:(1)dH(T)/dt=∇(λ(T)∇T)
where *T* (°C) is the temperature, λ (W/m/°C) denotes the thermal conductivity and *H* (J) the specific enthalpy which can be defined as:(2)$$H\left(T\right)=\int_{T_{0}}^{T}\rho \left(\tau \right)C_{p}\left(\tau \right)d\tau +g_{1}\left(T\right)L_{\rho }\left(T_{s}\right)$$
where *T*_0_ (°C) is an arbitrary reference temperature, ρ (kg/m^3^) the density, *T_s_* (°C) the solidus temperature, *C_p_* (J/kg/°C) the specific heat, *g*_l_ the volume fraction of liquid, and *L* (J/kg) the specific latent heat of fusion. In the one-phase modelling, *g_s_ (T)* is calculated using the micro-segregation model.

The average heat convection applied on the free surface of the mesh of the metal can be defined as:(3)$$-\lambda \nabla Tn=h\left(T-T_{ext}\right)$$
where *h* (W/(m²∙°C)) is the heat transfer coefficient, and Text is the external temperature.

At the ingot/mold interface, heat transfer is characterized with a Fourier type equation:(4)$$-\lambda \nabla Tn=1/R_{eq}\left(T-T_{mold}\right)$$
where *T_mold_* is the interface temperature of the mold and *R_eq_* (W/m²/°C)^−1^ is the heat transfer resistance that depends on the air gap and/or the local normal stress, as presented below:*R_eq_* = 1/(*min*(1/*R*_0_,1/*R_air_* + 1/*R_rad_*)) + *R_s_*  *if**e_air_* > 0(5)
*R_eq_* = 1/(1/ *R_σ_* + 1/*R_0_*) + *R_s_*  *if**e_air_* = 0(6)
where $R_{\mathrm{air}}=e_{\mathrm{air}}/\lambda_{\mathrm{air}}$ and $R_{s}=es/\lambda s$ respectively are the air gap and an eventual other body (typically slag) thicknesses and *λ_air_* and *λ_s_* are the air and the eventual other body thermal conductivities. *R*_0_ is the nominal heat resistance depending on the surface roughness. $R\sigma =1/{A\sigma }_{n}^{m}$ is the heat resistance taking into account the normal stress $\sigma_{n}$, A and *m* are the parameters of the pressure-heat flow law.
(7)$$R_{rad}=\frac{1}{\varepsilon }+\frac{1}{\varepsilon_{mold}}-1/\sigma_{stef}\left(T^{2}+T_{mold}^{2}\right)\left(T+T_{mold}\right)$$
where $\varepsilon_{mold}$ is the emissivity of the mold.

Table 2 shows the simulation conditions used in this study.

Figure 2 shows the solidified ingots obtained with various hot-top heights. The numerical results showed that severe shrinkage porosities appeared in the ingot with the free hot top. On the contrary, the opposite was observed with an increased hot top height.

Figure 3 shows the total solidification time obtained with various hot-top heights. The solidification time decreased with the decreasing hot-top height. The reduced solidification time causes non-uniform solidification and shrinkage cavities in the ingot with the reduced hot-top height.

In the metal-casting industry, the Niyama criterion (*Ny*) is commonly regarded as the standard measure of the potential for feeding-related shrinkage porosities [35,36,37,38]. *Ny* is defined as the local thermal gradient divided by the square root of the local cooling rate. For a sufficiently large *Ny*, no shrinkage cavities form. 

In this study, the inverse value of the Niyama criterion (*Ny*^−1^) shown in Equation (8) was used to predict the possibility of solidification shrinkage defects in the steel castings.
(8)$$Ny^{-1}=\sqrt{\dot{T}}/G_{s}$$
where $\dot{T}$ (K/s) is the cooling rate, and $G_{s}$ (K/mm) is the thermal gradient. *Ny*^−1^ is a common output variable of simulations used by foundries. Shrinkage porosity is likely to occur if the calculated *Ny*^−1^ is above a critical value. 

Figure 4 shows the *Ny*^−1^ obtained with the various hot-top heights. As the *Ny*^−1^ increases so does the probability of shrinkage. As shown in the figure, the ingot with the free hot top (Figure 4c) exhibits the highest *Ny*^−1^ value, which implies a high probability of shrinkage cavity formation in the ingot.

### 2.2. Materials and Forging Experiments

As shown by the finite element simulations, the ingot with the free hot top shows the highest *Ny*^−1^ values and the highest probability of shrinkage cavity formation in the ingot. Therefore, the ingot with the free hot top was excluded from the experiments, i.e., the conventional ingot having a 16% weight percentage was compared with the ingot with a reduced hot top with an 8% weight percentage to investigate the effects of the hot-top height on the heavy ingot forgings.

The experiments were carried out on a large 34CrNiMo6 forging ingot that is used as the raw material for producing main shafts in wind power plants. Table 3 shows the chemical composition of 34CrNiMo6 used for the ingot casting.

A cast ingot made of 34CrNiMo6 and consisting of three zones (that are typical of cast structures, namely columnar, chill, and equiaxed structures) was heated to 1250 °C. The forging was performed in the temperature range of 950–1150 °C to ensure sound internal structure. In this study, the ingots that were deformed by the initial upsetting were plastically deformed by cogging and necking. Figure 5 shows the steps implemented to forge the shaft and bar products: ingot casting, ingot heating, upsetting, and cogging and necking. Moreover, a quenching–tempering treatment was performed to enhance the mechanical properties of the forged products. The accurate measurement and control of the temperature during the thermal processes are very important to enhance product quality and operational performance [39,40,41,42,43,44,45]. Figure 6 shows the heat-treatment cycle used for the quenching–tempering of the forged main shaft.

## 3. Results and Discussion

In order to produce high-quality heavy forging products, it is crucial to consider the characteristics of the ingot during the design of the forging process. To develop tailored forging schedules for the reduced ingot hot-top heights, various hot forging schedules have been implemented and compared.

### 3.1. Main Shaft Forging

A series of main shaft forging operations have been carried out on the ingots with 16% and 8% hot tops, as shown in Table 4. Forging reduction is generally considered to be the cross-sectional reduction that takes place during the forging of an ingot or bar. In this study, ‘3S’ refers to the ratio of the original cross-section and the final cross-section being 3. Figure 7 compares the two forging methods performed in this study. For method 1, which follows the conventional main shaft forging operations, the hot-top side exhibits a less significant forging reduction. On the contrary, method 2 has been designed to induce a more pronounced forging reduction at the hot-top side. Figure 8 shows the shop floor operations implemented in the current study. 

The internal voids in the forged main shafts have been examined by ultrasonic testing (UT). Table 5 shows the UT results of the forged main shafts. No defects were observed in Case 1, while Ø 2.0–4.2 mm internal void defects were observed at the flange zone of the main shaft in Case 2, where the ingot with the reduced 8 % hot top was used. Apparently, the reduced hot-top height increases the internal voids because the decreased volume and heat capacity of the hot top slows down the solidification and causes non-uniform solidification and the formation of shrinkage cavities in the ingot. Consequently, the forging should penetrate the ingot internally through the increased forging reduction in order to obtain a high-quality main shaft. Increased deformations in the hot-top zone of the ingot demonstrated that the internal voids in the ingot could be eliminated more efficiently due to severe deformations [46]. As the flange zone of the main shaft exhibits a 2.2S forging ratio, the probability of the complete removal of internal voids is not sufficiently high. In contrast to the conventional ingot forging applied in method 1, the hot-top side is located in the heavy forging zone, and thus a more pronounced reduction has been achieved by method 2. It can be clearly seen from Figure 7b that efficient work along the hot-top zone of the ingot can be achieved when method 2 is applied. As expected, Case 3, where the main shaft is forged by method 2, did not show any internal defects.

Hence, to effectively control the number of internal voids in the ingot with reduced hot-top heights, and hence, to produce high-quality main shafts, method 2 is suggested. Ultimately, reaching a forging reduction ratio above 3S is recommended for the entire ingot as a novel criterion for the reduced hot-top design. This practical criterion ensures that the number of internal voids has been greatly reduced through the increased forging reduction of the hot-top zone.

### 3.2. Round Bar Forging

The above-mentioned forging strategy has been utilized to round bar forging. Figure 9 schematically shows the round bar forging process. Figure 10 shows the shop floor operations implemented in the current study. As the number of internal voids was reduced by increasing the forging reduction at the hot-top zone, forging ratios of 3S, 4S, and 5S were implemented, as shown in Figure 11.

The internal voids in the forged round bars were examined by UT. Table 6 shows the UT results of the forged round bars. No defects were observed with reduction ratios above 4S. Meanwhile, with 3S, Ø 1.5–3.8 mm internal void defects were observed at the central parts of the hot-top zone. By increasing the forging ratio, the number of forging hits also increases. Consequently, the internal voids are closed more effectively. Hence, to efficiently control the number of internal voids in ingots with reduced hot-top heights, the use of an increased forging reduction ratio is suggested. The practical criterion for the minimum forging reduction ratio depends on the product geometry, the permissible number of defects, and processing parameters of the forging process.

### 3.3. Mechanical Properties

To estimate the effects of the reduced hot-top height on their mechanical properties, the tensile properties of the forged main shafts with 16% and 8% hot tops were measured after the quenching–tempering heat treatment. Figure 12 shows the positions from which specimens were taken. As shown in the figure, specimens were taken from the zone with a forging ratio of 4.5S. Specimens for tensile testing were prepared in accordance with ASTM A370 “Standard Test Methods and Definitions for Mechanical Testing of Steel Products” [47].

Figure 13 shows the results of the microscopic analysis. Tempered martensite is the phase mainly observed in both main shafts. Table 7 compares the mechanical properties of the two main shafts. The mechanical properties of the two main shafts after quenching-tempering were similar.

Thus, it can be concluded that strategic forging schedules are required to overcome the relatively inferior ingot microstructure induced by the reduced hot-top heights. To improve the utilization ratio of ingots, the ingot hot-top height must be minimized. To compensate for the deteriorating effects of the reduced hot-top heights, an increased forging reduction ratio at the hot-top zone is strongly recommended. 

To estimate the mechanical properties of the forged bars, tensile tests have been performed. As the mechanical properties of forged bars are directly influenced by the forging ratio regardless of hot-top height, the mechanical properties of the forged bars were measured for forged bars with 8 % hot-top at forging ratios of 3S, 4S, and 5S. Figure 14 shows the positions from which specimens were taken. Table 8 compares the mechanical properties of the forged bars at various forging ratios. As can be expected, the mechanical properties of the forged bars increase with increasing forging ratio.

## 4. Summary and Future Research

For cost-effective hot forging of heavy ingots, the size of the hot top should be decreased while increasing the forging reduction ratio. As the quality of heavy ingots is commonly affected by internal voids, extensive studies on optimum hot-top heights and forging schedules must be performed to efficiently and effectively control the number and size of any internal voids. From the operational results, the present work successfully implements effective forging steps to produce acceptable shaft and bar products using ingots with reduced hot tops. The design of appropriate forging sequences that increase the reduction ratio of hot tops proved to be practical and feasible. 

The ultimate application goal of the reduced hot-top height for large ingot casting is to realize the process design and optimization by improving soundness and homogeneity (both structural and compositional) of the ingot, minimizing the preparation and material costs, reducing energy loss, etc. As the past modeling/simulation experiences demonstrate the practical and potential significance, studies on the numerical simulation program and the input data will be continued to achieve optimum process design. Understanding the formation and closure mechanisms of shrinkage porosity is still an active topic of research. 

## 5. Conclusions

Feasibility studies to investigate decreasing ingot hot-top heights for cost-effective hot forging of heavy ingots were performed. The following conclusions were obtained:(1)For cost-effective hot forging of heavy ingots, the size of the hot top should be decreased while increasing the forging reduction ratio.(2)As the quality of heavy ingots is commonly affected by internal voids, optimum hot-top heights and forging schedules must be determined to efficiently control the number and size of the internal voids.(3)The design of appropriate forging sequences that increase the reduction ratio of hot tops can enhance the void closure in the hot-top zone. Therefore, it is favorable to consider the forging geometry when designing the reduced hot top of heavy ingots.(4)According to the operational results implemented in the present work, the forging reduction ratio in the hot-top zone must be maximized. To produce acceptable shaft and bar products using ingots with reduced hot-top heights, forging reduction ratios of at least 3S and 4S are required, respectively.(5)The mechanical properties of shop floor products created from ingots with reduced hot-top heights are not inferior and are similar to those of conventional ingots with larger hot tops.

## Figures and Tables

**Figure 1 materials-13-02916-f001:**
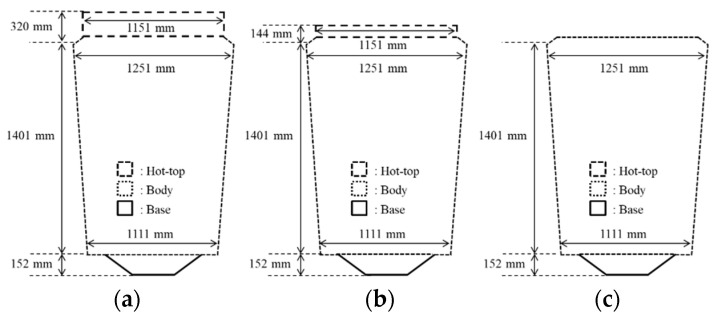
Schematic drawing of the ingot dimensions: (**a**) 16% hot top; (**b**) 8% hot top; (**c**) free hot top.

**Figure 2 materials-13-02916-f002:**
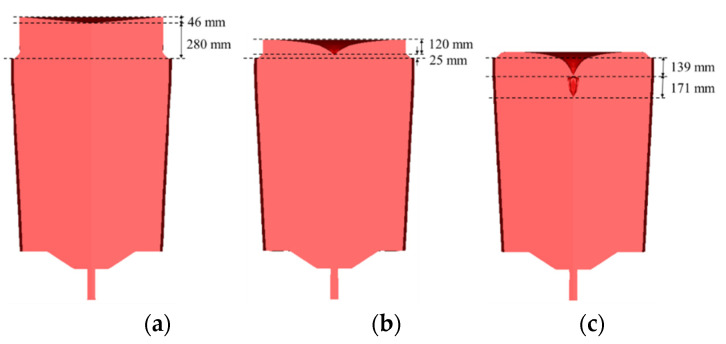
Solidified ingots obtained for (**a**) 16% hot top; (**b**) 8% hot top; (**c**) free hot top.

**Figure 3 materials-13-02916-f003:**
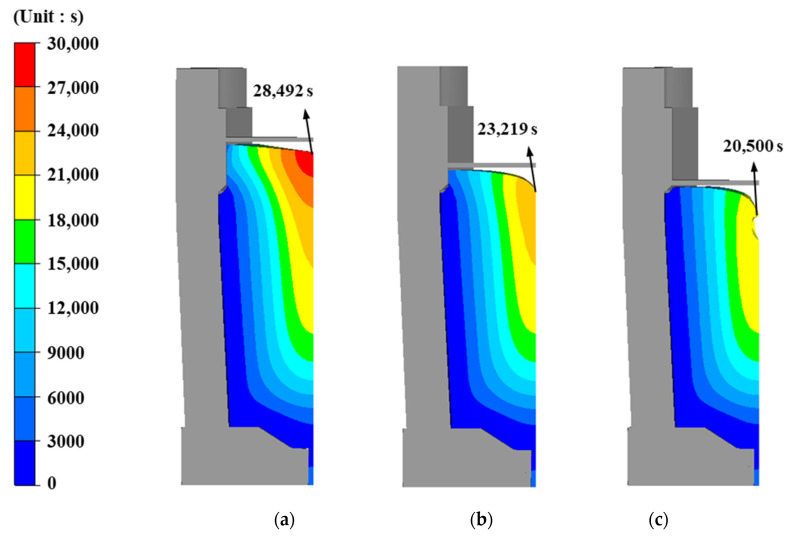
Total solidification time for (**a**) 16% hot top; (**b**) 8% hot top; (**c**) free hot top.

**Figure 4 materials-13-02916-f004:**
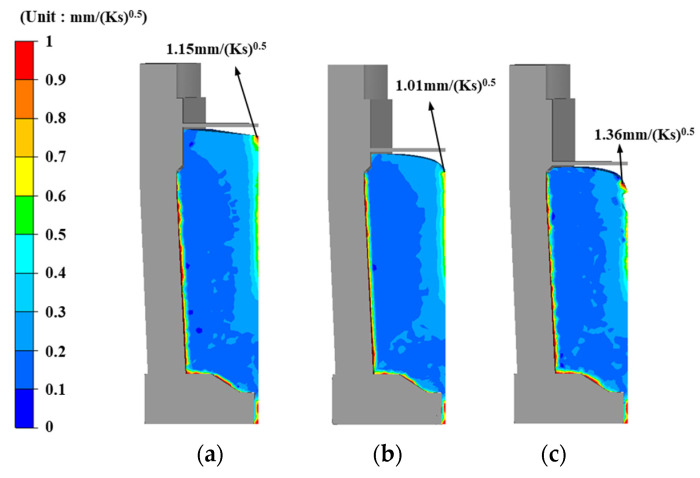
*Ny*^−1^ for (**a**) 16% hot top; (**b**) 8% hot top; (**c**) free hot top.

**Figure 5 materials-13-02916-f005:**
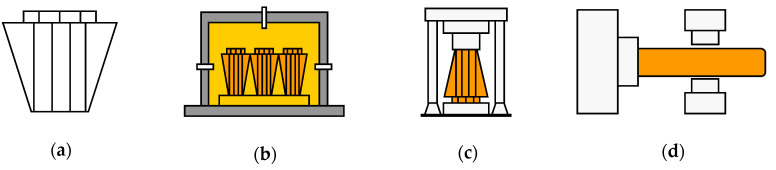
Schematic drawing of the main shaft forging process: (**a**) ingot; (**b**) heating; (**c**) upsetting; (**d**) cogging and necking.

**Figure 6 materials-13-02916-f006:**
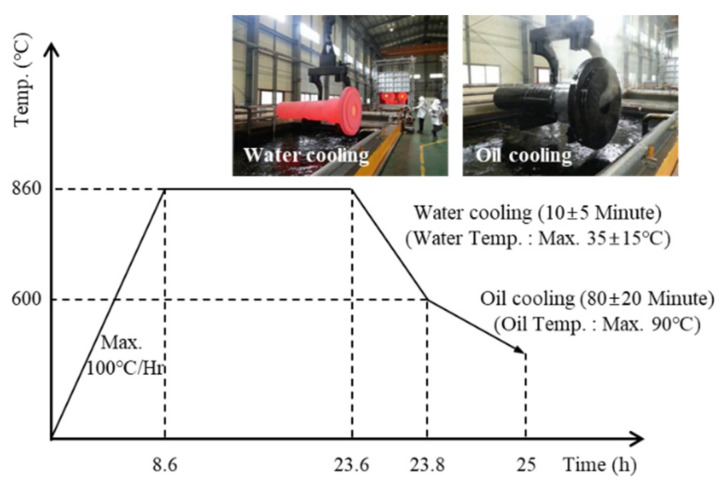
Heat treatment cycle for the quenching–tempering process of the forged main shaft.

**Figure 7 materials-13-02916-f007:**
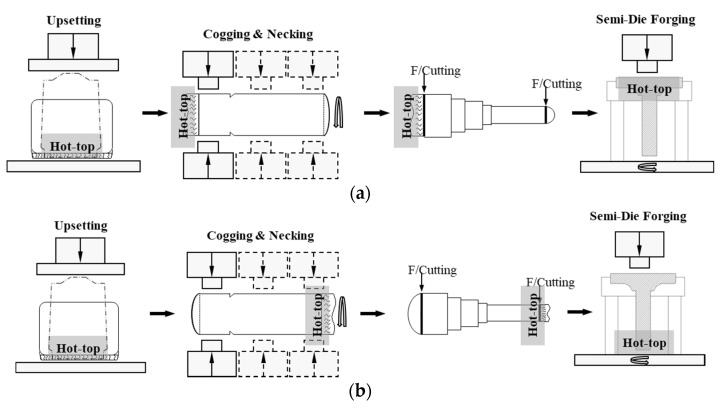
Schematic drawing of the main shaft forging: (**a**) Method 1; (**b**) Method 2.

**Figure 8 materials-13-02916-f008:**
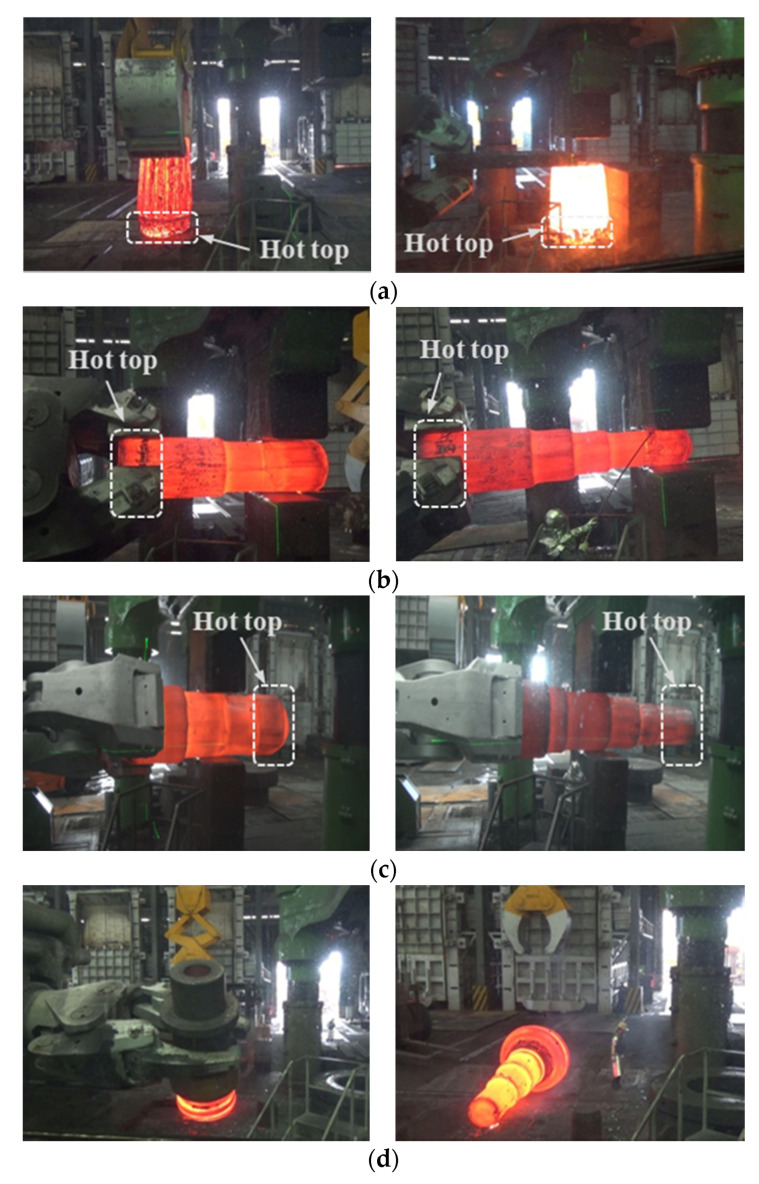
Shop floor main shaft forging: (**a**) upsetting; (**b**) cogging and necking for method 1; (**c**) cogging and necking for method 2; (**d**) semi-die forging.

**Figure 9 materials-13-02916-f009:**
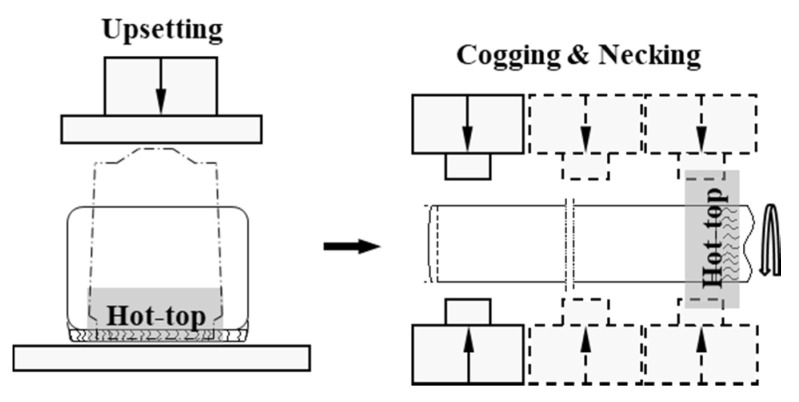
Schematic drawing of the round bar forging.

**Figure 10 materials-13-02916-f010:**
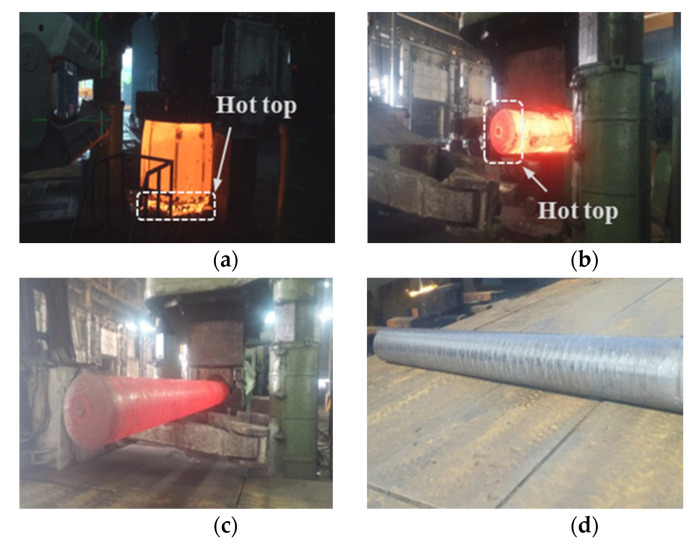
Shop floor round bar forging: (**a**) upsetting; (**b**) bar cogging; (**c**) round bar forging; (**d**) forged bar.

**Figure 11 materials-13-02916-f011:**
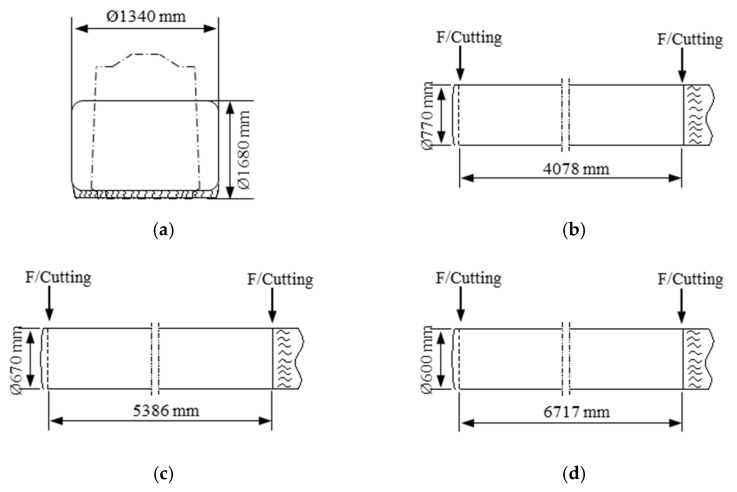
Dimension of the forged round bar: (**a**) ingot with the 8% hot top; (**b**) 3S; (**c**) 4S; (**d**) 5S.

**Figure 12 materials-13-02916-f012:**
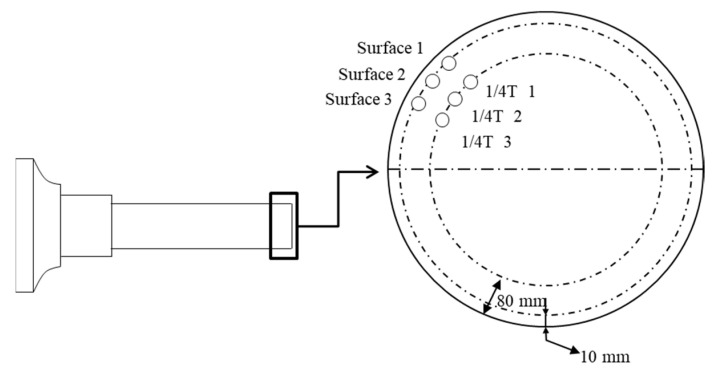
Test specimen extraction positions.

**Figure 13 materials-13-02916-f013:**
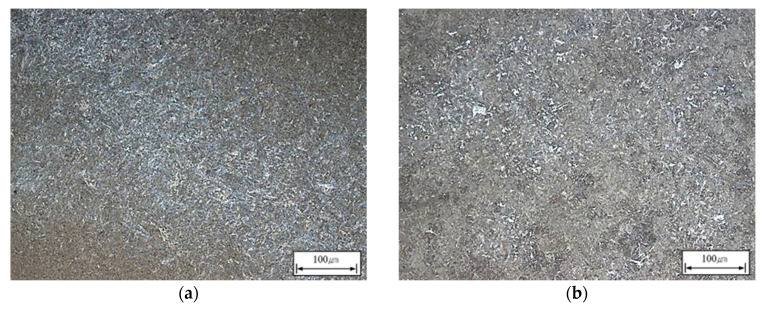
Microstructures at position 1/4T of the (**a**) 16% hot top; (**b**) 8% hot top.

**Figure 14 materials-13-02916-f014:**
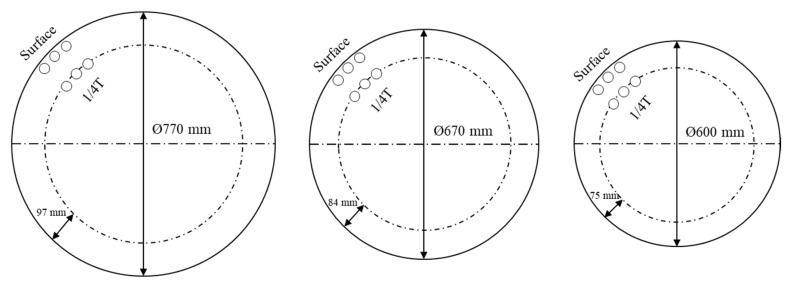
Test specimen extraction positions.

**Table 1 materials-13-02916-t001:** Weights of the ingots.

Ingot Type	Weight (kg)			
	Total	Body	Hot Top	Base
16% hot top	16,100	13,324	2576	200
8% hot top	14,683	13,324	1159	200
Free hot top	13,524	13,324	0	200

**Table 2 materials-13-02916-t002:** Simulation conditions for ingot casting.

Condition	Value (Unit)
Ingot material	34CrNiMo6
Mold material	FC200 (3.8C-1.0Si-0.4Mn)
Computation mode	Thermomechanical mode
Analysis model	1/8 model
Ingot body weight	13,324 [kg]
Superheat	30 (°C)
Element type	Tetrahedron
Element number	34,526

**Table 3 materials-13-02916-t003:** Chemical composition of ingot used for ingot casting, wt%.

Fe	C	Si	Mn	P	S	Cu	Ni	Cr	Mo	Al	V
Bal	0.33	0.31	0.60	0.015	0.003	0.07	1.59	1.70	0.18	0.026	0.047

**Table 4 materials-13-02916-t004:** Experimental conditions for main shaft forging.

Test No.	Forging Process	Hot-Top Type	Forging Ratio
Case 1	Method 1	16% hot top	2.2S–4.5S
Case 2	Method 1	8% hot top	2.2S–4.5S
Case 3	Method 2	8% hot top	2.2S–4.5S

**Table 5 materials-13-02916-t005:** UT results of the forged main shafts.

No.	2.2S	3.1S	3.8S	4.5S
Case 1	No Indication	No Indication	No Indication	No Indication
Case 2	Defects in flange zone	No Indication	No Indication	No Indication
Case 3	No Indication	No Indication	No Indication	No Indication

**Table 6 materials-13-02916-t006:** Result of the round bar forging.

Forging Ratio	Internal Defects	Forging Hit Number
3S	Defects	1080
4S	No Indication	1260
5S	No Indication	1440

**Table 7 materials-13-02916-t007:** Mechanical properties of the main shafts after the quenching–tempering heat treatment.

Hot-Top Type	Test Number	Position	Tensile Properties			
			Yield Strength (MPa)	Tensile Strength (MPa)	Elongation	Reduction of Area
16% Hot-top Ingot (Case 1, method 1)	1	Surface	808.8	918.1	21.0	64.2
		1/4T	772.5	892.2	21.4	65.4
	2	Surface	806.7	915.0	21.3	64.1
		1/4T	770.5	889.0	21.5	65.6
	3	Surface	812.2	922.6	21.1	64.1
		1/4T	780.3	897.4	21.9	65.6
	Average	Surface	809.2	918.6	21.1	64.1
		1/4T	774.4	892.9	21.6	65.5
8% Hot-top Ingot (Case 3, method 2)	1	Surface	809.0	915.4	21.0	63.9
		1/4T	780.9	899.1	21.4	65.0
	2	Surface	810.9	920.2	20.8	63.7
		1/4T	777.5	888.8	21.5	65.7
	3	Surface	803.7	912.4	21.4	64.2
		1/4T	780.5	903.4	21.7	64.7
	Average	Surface	807.9	916.0	21.1	63.9
		1/4T	779.6	897.1	21.5	65.1

**Table 8 materials-13-02916-t008:** Mechanical properties of the forged bars at various forging ratios

Forging Ratio	Position	Tensile Properties			
		Yield Strength (MPa)	Tensile Strength (MPa)	Elongation	Reduction of Area
3S	Surface	591	722	18	60
	1/4T	582	711	17	60
4S	Surface	608	739	19	61
	1/4T	601	733	19	60
5S	Surface	621	760	20	65
	1/4T	617	754	20	64

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
