# Peer review of "Feasibility of Reduced Ingot Hot-Top Height for the Cost-Effective Forging of Heavy Steel Ingots"

_materials, 2020, doi:10.3390/ma13132916_

Round 1
Reviewer 1 Report
The work concerns the development of efficient ingot forging technology that allows reducing the hot-top ingot in order to obtain a product without internal porosity (voids). In the first place, numerical modeling was carried out for conventional ingot and ingots with 16 and 8% sprue. Then, based on the FEM results obtained, due to the high probability of voids, ingots with springs were selected for further research, for which two shaft manufacturing technologies were proposed (3 cases in total). As expected, much better results were obtained for technology in which much larger deformations took place in the head, regardless of its size. Then this technology was used to determine the size of the forging degree during the production of bars. In this case, it could also be predicted that the best results would be achieved for the highest degree of forging.
The high application and utilitarian potential of the research results presented in the manuscript should be emphasized.
Of the 35 literature references cited by the authors of the manuscript, the vast majority are of Asian origin (of course, without compromising the scientific quality of these works); it would be appreciated to consider adding / exchanging for other positions of non-Asian authors.
Distance between lines 70-74 in paragraph is different (to big).
I suggest to remove information from lines 70-80 into chapter 1. A new chapter on Research Methodology / Aim of the manuscript should appear from line 81, because form this line is started description of Authors research. Or current chapter 3. Materials and forging experiments, sholud be removed before line 81, as a main chapter and numerical modelling should be as a subchapter in this chapter, I mean 3. Materials and forging experiments.
It would be useful to have more information on the boundary-initial conditions taken in numerical modeling and other information on the results of the simulation.
The description in lines 117-125 shows that as the Niyama criterion (Ny-1) increases, the likelihood of defects increases. So why, for 16% hot-top (Fig. 4a) is this parameter higher than for 8% hot-top (Fig. 4b), since Total solidification time is the smallest for free hot-top?
Why in the manuscript the authors base only on the inverted Niyama criterion and not e.g. Cockcroft-Latham or Freudenthal criteria regarding fracture models often used to determine voids in ingots?
Line 132. ingot Therefore
Line 145. 950–1,150 ℃
The pictures presented in Fig. 8b and 8c show no difference for both methods. What's more, the photos in Fig. 8a and 8b are the same. For what method is the picture shown (on the left) in Fig. 8d. I understand that the pictures show the process for the 8% hot-top not for 16 hot-top ?
I understand that in chapter 4.3 the mechanical properties were determined for shaft and not bar products? It seems that the work system is a bit logical, because, in chapter 4.1, the results for forging shafts are presented, in chapter 4.2 for forging bars, and finally in chapter 4.3 mechanical tests for shafts. Perhaps it is worth considering a better logic in the manuscript.
In addition, chapter 4.3 does not specify for which method (No. 1 or No. 2) forging of rollers elements for testing mechanical properties and for which degree of forging were selected.
After considering the comments and suggestions of the reviewer, the manuscript may be subject to further evaluation.
Author Response
Please refer to attached revision note.

Reviewer 2 Report
The paper analyzes the feasibility of reduced ingot hot-top height for the cost-effective forging of heavy steel ingots.
From the analysis of the information presented in the article, I found the following:
- the last two keywords are not representative for the content of the paper;
- the statement from lines 36-39 is a very general one, although it refers to many bibliographic sources and in these conditions it should be detailed;
- the research methodology is unclear and needs to be substantially improved;
- finite element analysis (FEM) is not relevant because the results obtained are known in practice;
- the temperature at which the cooling media is changed must be specified (Fig. 6). Also, the time values in Figure 6 must be completed;
- images with ultrasound test results must be presented;
- the resolution of Figure 13 needs to be substantially improved;
- the decision by which only the shafts type parts and not the round bar type parts were subjected to mechanical tests is not justified.
- images with the test tubes used in the tests must be presented;
- the discussion part is completely missing and the novelty in the field brought by the research presented in the paper cannot be identified.
- future research directions are not presented;
Thus, the article cannot be accepted for publication in this form. Substantial changes to the article are required so that it can be published.
Author Response
Please refer to the attached revision note.

Round 2
Reviewer 1 Report
Dear Authors,
The authors introduced and included most of the suggestions and comments from the reviews. Therefore, I believe that in its current form, the article can be published.
Reviewer 2 Report
The authors responded to the made comments in review. The article can be published in the presented form.